# Therapeutic Effect of Mitochondrial Division Inhibitor-1 (Mdivi-1) on Hyperglycemia-Exacerbated Early and Delayed Brain Injuries after Experimental Subarachnoid Hemorrhage

**DOI:** 10.3390/ijms23136924

**Published:** 2022-06-22

**Authors:** Chia-Li Chung, Yu-Hua Huang, Chien-Ju Lin, Yoon-Bin Chong, Shu-Chuan Wu, Chee-Yin Chai, Hung-Pei Tsai, Aij-Lie Kwan

**Affiliations:** 1Graduate Institute of Medicine, College of Medicine, Kaohsiung Medical University, Kaohsiung 807, Taiwan; r1chung@yahoo.com.tw; 2Department of Surgery, Kaohsiung Municipal Siaogang Hospital, Kaohsiung 81267, Taiwan; 3Department of Neurosurgery, Kaohsiung Chang Gung Memorial Hospital and Chang Gung University College of Medicine, Kaohsiung 833401, Taiwan; newlupin2001@yahoo.com.tw; 4School of Pharmacy, College of Pharmacy, Kaohsiung Medical University, Kaohsiung 807, Taiwan; mistylin@kmu.edu.tw; 5Division of Neurosurgery, Department of Surgery, Kaohsiung Medical University Hospital, Kaohsiung 80756, Taiwan; bin99068@hotmail.com (Y.-B.C.); shuchuan623@gmail.com (S.-C.W.); 6Department of Pathology, Kaohsiung Medical University Hospital, Kaohsiung 807, Taiwan; ccjtsai@yahoo.com; 7Department of Pathology, College of Medicine, Kaohsiung Medical University, Kaohsiung 807, Taiwan; 8Department of Surgery, College of Medicine, Kaohsiung Medical University, Kaohsiung 807, Taiwan; 9Department of Neurosurgery, University of Virginia, Charlottesville, VA 22903, USA

**Keywords:** hyperglycemia, Mdivi-1, SAH

## Abstract

Background: Neurological deficits following subarachnoid hemorrhage (SAH) are caused by early or delayed brain injuries. Our previous studies have demonstrated that hyperglycemia induces profound neuronal apoptosis of the cerebral cortex. Morphologically, we found that hyperglycemia exacerbated late vasospasm following SAH. Thus, our previous studies strongly suggest that post-SAH hyperglycemia is not only a response to primary insult, but also an aggravating factor for brain injuries. In addition, mitochondrial fusion and fission are vital to maintaining cellular functions. Current evidence also shows that the suppression of mitochondrial fission alleviates brain injuries after experimental SAH. Hence, this study aimed to determine the effects of mitochondrial dynamic modulation in hyperglycemia-related worse SAH neurological prognosis. Materials and methods: In vitro, we employed an enzyme-linked immunosorbent assay (ELISA) to detect the effect of mitochondrial division inhibitor-1 (Mdivi-1) on lipopolysaccharide (LPS)-induced BV-2 cells releasing inflammatory factors. In vivo, we produced hyperglycemic rats via intraperitoneal streptozotocin (STZ) injections. Hyperglycemia was confirmed using blood-glucose measurements (>300 mg/dL) 7 days after the STZ injection. The rodent model of SAH, in which fresh blood was instilled into the craniocervical junction, was used 7 days after STZ administration. We investigated the mechanism and effect of Mdivi-1, a selective inhibitor of dynamin-related protein (Drp1) to downregulate mitochondrial fission, on SAH-induced apoptosis in a hyperglycemic state, and evaluated the results in a dose–response manner. The rats were divided into the following five groups: (1) control, (2) SAH only, (3) Diabetes mellitus (DM) + SAH, (4) Mdivi-1 (0.24 mg/kg) + DM + SAH, and (5) Mdivi-1 (1.2 mg/kg) + DM + SAH. Results: In vitro, ELISA revealed that Mdivi-1 inhibited microglia from releasing inflammatory factors, such as tumor necrosis factor-α (TNF-α), interleukin (IL)-1β, and IL-6. In vivo, neurological outcomes in the high-dose (1.2 mg/kg) Mdivi-1 treatment group were significantly reduced compared with the SAH and DM + SAH groups. Furthermore, immunofluorescence staining and ELISA revealed that a high dose of Mdivi-1 had attenuated inflammation and neuron cell apoptosis by inhibiting Hyperglycemia-aggravated activation, as well as microglia and astrocyte proliferation, following SAH. Conclusion: Mdivi-1, a Drp-1 inhibitor, attenuates cerebral vasospasm, poor neurological outcomes, inflammation, and neuron cell apoptosis following SAH + hyperglycemia.

## 1. Introduction

Spontaneous subarachnoid hemorrhage (SAH) is a devastating disease with high mortality and morbidity rates [1]. Only one-third of survival-to-discharge patients resume the same employment as pre-event [2]. Even patients with favorable outcomes are frequently left with significant impairment to residual memory or executive function, or language deficits [3]. Early or delayed brain injuries, presenting in the first 72 h or later within 14 days following SAH, are major components associated with neurological sequelae [4,5,6].

One of the critical mechanisms of early brain injury is the disruption of the blood–brain barrier (BBB), a structure that primarily consist of cerebral microvascular endothelial cells, astrocytic endfeet, an extracellular matrix, and pericytes [7]. Loss of BBB integrity results in the direct exposure of the brain tissues to neurotoxic blood contents and immune cells, which leads to secondary brain insults, including inflammation and oxidative stress, as well as other cascades [8]. In addition, apoptosis is one major catastrophic event in the early stage of SAH [6,9]. Apoptotic cell death, which can occur in cerebral neurons or endothelial cells through the intrinsic or extrinsic pathways, has played a significant role in the prognosis in animal SAH models [4,10,11]. Meanwhile, the most common cause of delayed brain injuries is cerebral arterial vasospasm. Cerebral ischemia secondary to vasospasm occurs in 20–30% of these patients, and has been correlated with a 1.5–3-fold increase in mortality in the first 2 weeks following SAH [12,13].

Mitochondrial activity modulation has been reported to alleviate brain injuries and improve neurological deficits following experimental SAH [14,15]. Mitochondria are among the irreplaceable endomembrane systems and are highly dynamic organelles that constantly fuse and divide [16]. Mitochondrial fusion and fission are important to maintain functions, including energy provision to cells, anabolic and catabolic biochemical pathway intervention, calcium homeostasis, regulation, and cell-death initiation [17,18]. The morphology or the shape change in mitochondria is mediated mainly by dynamin-related protein (Drp1) and mitochondrial fission 1 (Fis1) for fission, and mitofusin (Mfn) and optic atrophy-1 (OPA1) for fusion [19]. Defects in mitochondrial fission and fusion proteins have been linked to neurodegenerative diseases such as Alzheimer’s, Parkinson’s, and Huntington’s disease, with the essential role of mitochondrial dynamics in neurons [20]. In addition, the loss of balance in fusion and fission leads to acute central nervous system injuries. The studies demonstrated that Drp1 inhibition could attenuate the neurological dysfunction of traumatic brain injury or spinal cord injury by inhibiting mitochondrial fragmentation and apoptosis activation [21,22]. Drp1 downregulation or Mfn2 upregulation reduces neuronal apoptosis by restoring mitochondrial function in hypoxic or ischemic brain models [23,24].

It has been reported that hyperglycemia after cerebral ischemia, a known detrimental factor, further exacerbates the imbalance between mitochondrial fission and fusion, and favors mitochondrial fragmentation and subsequent mitochondrial damage [25]. Our previous study demonstrated that hyperglycemia was partly involved in early brain injuries in as SAH rat model, and induced more profound apoptosis, which mostly occurs in the neurons of the cerebral cortex. Furthermore, morphologically, we found that hyperglycemia exacerbated post-SAH vasospasm, as evidenced by the greater cross-sectional area reduction of the basilar arteries.

Drp1 is demonstrated as an intrinsic component of multiple mitochondria-dependent apoptosis pathways, and its activity predominantly controls mitochondrial fission [26]. Drp1 proteins, mainly in the cytoplasm, translocate to the mitochondria and are subjected to several post-transcriptional modifications, including ubiquitination, phosphorylation, nitrosylation and SUMOylation [27,28]. Mitochondrial division inhibitor-1 (Mdivi-1) is a quinazolinone derivative that can selectively inhibit Drp1 to downregulate mitochondrial fission [29]. Investigations have reported that Mdivi-1 helps attenuate cell death following experimental traumatic brain injury by maintaining mitochondrial morphology, mitigating autophagy dysfunction, or inhibiting apoptosis activation [21,30]. This study investigated the influence of Mdivi-1 on cell death, severe neurological outcomes, and neuronal apoptosis following SAH with hyperglycemia.

## 2. Results

### 2.1. ELISA Assay for Inflammatory Factors In Vitro

The TNF-α, IL-1β, and IL-6 levels in the supernatant samples from BV-2 were examined using ELISA 6 h after LPS induction to investigate the relationship between pro-inflammatory factors and Mdivi-1. ELISA revealed that LPS induced the release of TNF-α, IL-1β, and IL-6; however, Mdivi-1 blocked the release of TNF-α (Figure 1A), IL-1β (Figure 1B), and IL-6 (Figure 1C) after LPS induction.

### 2.2. Neurological Outcomes

The neurobehavioral scores, including ambulation, placing/stepping reflex, and MDI, were not different between the SAH, DM + SAH, and DM + SAH + low-dose Mdivi-1 (0.24 mg/kg) groups (Table 1). In animals subjected to SAH, both the ambulation (1.91 ± 0.37) and placing/stepping reflex (1.34 ± 0.11) scores were significantly lower in the SAH animals than in the DM + SAH animals (ambulation: 2.39 ± 0.27 and placing/stepping reflex: 1.77 ± 0.19). Treatment with a high dose of Mdivi-1 (1.2 mg/kg) significantly decreased both the ambulation (1.62 ± 0.13; *p* < 0.05) and the placing/stepping reflex (0.57 ± 0.21) scores when compared with the DM + SAH group, but treatment with the low-dose Mdivi-1 (0.24 mg/kg) did not (ambulation: 2.11 ± 0.19 and placing/stepping reflex: 1.59 ± 0.29). Similarly, the MDI in the high-dose Mdivi-1 (1.2 mg/kg) treatment group (1.89 ± 0.31; *p* < 0.05) was also significantly reduced when compared with that in the DM + SAH group (4.20 ± 0.31) (Table 1).

### 2.3. Morphological, Cross-Sectional-Area, and Thickness Changes in Basal Artery (BA)

Microscopic examination revealed endothelial deformation, internal elastic laminae twisting, and smooth muscle necrosis in the BAs of rats subjected to normal, SAH, DM + SAH, DM + SAH + low-dose Mdivi-1, and DM + SAH + high-dose Mdivi-1 (Figure 2A). The mean cross-sectional areas of BA were 0.52 ± 0.11, 0.21 ± 0.059, 0.14 ± 0.028, 0.23 ± 0.051, and 0.37 ± 0.060 mm^2^ in the control, SAH, DM + SAH, DM + SAH + low-dose Mdivi-1 (0.24 mg/kg), and DM + SAH + high-dose Mdivi-1 (1.2 mg/kg) groups, respectively (Figure 2B). The BA cross-sectional area in the DM + SAH + high-dose Mdivi-1 (1.2 mg/kg) group was reduced by 37.8% (*p* < 0.05) and 47.0% (*p* < 0.001) compared with that in the SAH and DM + SAH group, respectively. The BA thickness exhibited no significant difference between the SAH (0.0328 ± 0.006 mm) and DM + SAH only (0.0333 ± 0.005 mm) groups (Figure 2C). A significant increase in the BA thickness was observed in the DM + SAH + high-dose Mdivi-1 (1.2 mg/kg) group (0.0183 ± 0.003 mm; *p* < 0.01 vs. both SAH and DM + SAH groups) (Figure 2C).

### 2.4. Microglia and Astrocyte Proliferation

Aside from the bleeding area, the activated microglia diffused into the brain parenchyma such as the brain stem, cortex, and hippocampus [31,32]. Similarly, after SAH, astrocytes were activated as part of gliosis [33]. This study employed immunofluorescence staining for Iba-1 and GFAP to detect the presence of microglial cells and astrocytes, respectively. Immunofluorescence staining for Iba-1 indicated that SAH induced microglial cell proliferation in the rat brain and was enhanced by hyperglycemia (Figure 3). In addition, quantitative analysis of the Iba-1 staining intensity revealed comparable levels between the control (set at 1.0), SAH (7.80 ± 1.88), DM + SAH (14.76 ± 2.07), DM + SAH + low-dose Mdivi-1 (0.24 mg/kg) (3.81 ± 1.80), and DM + SAH + high-dose Mdivi-1 (1.2 mg/kg) groups (2.15 ± 0.74). In contrast, Iba-1 staining in the brain of the SAH and DM + SAH rats was significantly elevated, and the Mdivi-1 treatment significantly reduced microglial cell proliferation (low-dose Mdivi-1: *p* < 0.01 compared with the SAH group and *p* <0.001 compared with the DM +SAH; high-dose Mdivi-1: *p* < 0.001 compared with the SAH group and P <0.001 compared with the DM + SAH group) (Figure 3B). The immunofluorescence staining results for GFAP were consistent with those of the Iba-1 staining. In addition, the quantitative analysis the of GFAP staining intensity revealed comparable levels between the control (set at 1.0), SAH (2.33 ± 0.72), DM + SAH (6.53 ± 1.32), DM + SAH + low-dose Mdivi-1 (0.24 mg/kg) (3.14 ± 1.63), and DM + SAH + high-dose Mdivi-1 (1.2 mg/kg) groups (1.63 ± 0.42). In contrast, GFAP staining in the brains of the SAH and DM + SAH rats was substantially elevated, and the Mdivi-1 treatment significantly reduced astrocyte proliferation (low-dose Mdivi-1: *p* < 0.001 compared with DM + SAH; high-dose Mdivi-1: *p* < 0.001 compared with DM + SAH) (Figure 4B).

### 2.5. Real-Time PCR of Pro-Inflammatory Factors

The TNF-α, IL-1β, and IL-6 levels in the brain were examined using real-time PCR 7 days after SAH, to investigate the relationship between pro-inflammatory factors in SAH and Mdivi-1. The real-time PCR result indicated that the TNF-α, IL-1β, and IL-6 expression in the DM + SAH group were significantly higher than those in the SAH group in the brain (Figure 5). All of the aforementioned increases in protein expression were significantly attenuated after the administration of the Mdivi-1 treatment (TNF-α: *p* < 0.05; IL-1β: *p* < 0.01; IL-6: *p* < 0.05, compared with the respective DM + SAH group upon treatment with low-dose Mdivi-1) (TNF-α, IL-1β, and IL-6: *p* < 0.001, compared with the respective DM + SAH group upon treatment with high-dose Mdivi-1) (Figure 5).

### 2.6. Apoptosis of Neuron Cells

This study conducted immunofluorescence staining for TUNEL and neuron cells to detect neuron cell apoptosis. Quantitative analysis of the number of instances of double immunofluorescence positive staining for TUNEL and NeuN revealed that SAH induced neuron cell apoptosis in the rat brain (Figure 6A). In contrast, double staining in the brain of hyperglycemic rats was substantially elevated (26.625 ± 9.50) and the Mdivi-1 treatment significantly reduced neuron cell apoptosis (low-dose Mdivi-1: 17.5 ± 3.93, *p* < 0.05; high-dose Mdivi-1: 8.125 ± 5.11, *p* < 0.001 compared with the DM + SAH group) (Figure 6B).

### 2.7. Western Blot Analysis

Drp1 is demonstrated as an intrinsic component of multiple mitochondria-dependent apoptosis pathways, and its activity predominantly controls mitochondrial fission [26]. Mdivi-1 is a derivative of quinazolinone that can selectively inhibit Drp1 to downregulate mitochondrial fission [29]. This study demonstrates that the phospho-Drp1 (p-Drp1) level in the DM + SAH group was significantly higher than that in the SAH group in the Western blot analysis. High-dose Mdivi-1 (1.2 mg/kg) treatment significantly reduced the p-Drp1 expression (*p* < 0.001). However, low-dose Mdivi-1 (0.24 mg/kg) had a trend toward decreasing the expression of p-Drp1, albeit not significantly (Figure 7).

## 3. Discussion

Our previous study demonstrated that hyperglycemia aggravated neuronal apoptosis that was related to unfavorable neurological outcomes following SAH in a rodent model. Enhancement of the extrinsic caspase cascade activation through the ERK signal pathway may be the mechanism underlying hyperglycemia-mediated apoptosis. Meanwhile, hyperglycemia exacerbated cerebral vasospasm and was associated with poor neurological outcomes after SAH. An increasing number of clinical studies have reported a correlation between mitochondrial fission/fusion and unfavorable prognosis in SAH cases. Fan et al. demonstrated that mitochondrial fission might inhibit mitochondrial complex I to become a cause of oxidative stress in SAH; moreover, they demonstrated that Drp1 inhibition by Mdivi-1 attenuated early brain injury following SAH, probably by suppressing inflammation-related BBB disruption and endoplasmic reticulum stress-induced apoptosis [15]. The neurological outcomes of our study revealed significantly reduced MDI in the high-dose (1.2 mg/kg) Mdivi-1 treatment group compared with the SAH and DM + SAH groups, but not in the low-dose Midiv-1 group. Morphologically, we previously revealed that hyperglycemia exacerbated post-SAH vasospasm, as evidenced by a significant reduction in the rat BA cross-sectional area and basilar artery thickness. In this study, treatment with a high-dose Mdivi-1 (1.2 mg/kg) significantly attenuated post-SAH vasospasm with or without hyperglycemia.

Many reports have revealed that glial cells (microglia and astrocyte) regulate SAH-induced vasospasm and neuronal apoptosis by releasing inflammation factors such as TNF-α, IL-1β, and IL-6. Inflammation and cytokines may participate in the pathology of BBB disruption and brain edema, which are characteristic features of both clinical and experimental SAH [34,35]. A variety of inflammatory cytokines, including IL-1β, IL-6, and TNF-α, are strongly associated with rat brain injury [36]. Neuronal apoptosis was inhibited in various experimental animal models of neurological disease, in addition to the anti-inflammatory effects. The systemic levels of TNF-α and IL-6 are elevated in DM, and can directly promote insulin resistance [37,38]. Thus, elevated cytokine levels may not only serve as DM markers, but also play a significant role in type 2 diabetes etiology. The tendency of diabetes patients to have higher levels of inflammation has serious consequences [39]. Microglia and astrocyte activation aggravates SAH-induced brain injury by secreting inflammatory factors [40,41,42], whereas the inhibition of microglia and astrocyte activation attenuates brain injury following SAH. Our results indicate that hyperglycemia enhanced SAH-induced microglia and astrocyte activation and proliferation, thereby increasing the inflammatory factor concentration in the CSF and the number of instances of neuron cell apoptosis in the brain. Mitochondrial fusion and fission are essential for maintaining mitochondrial functions, including energy metabolism, free-radical formation regulation, calcium circulation, and cell-death pathway initiation [43]. Previous studies demonstrated that Mdivi-1, a Drp-1 inhibitor, can provide neuroprotection against transient ischemic brain damage in vivo, and reduce the infarct volume [23]. In animals with SAH, the current evidence shows that Mdivi-1 potentially suppresses BBB disruption, oxidative stress, and endoplasmic-reticulum-stress-induced apoptosis [14,15]. Our study demonstrates that a high-dose of Mdivi-1 attenuated inflammation and neuron cell apoptosis by inhibiting SAH-induced activation and microglia and astrocyte proliferation, with or without hyperglycemia.

## 4. Materials and Methods

### 4.1. Cell Culture and Treatment

The mouse BV2 microglial cells were cultured with DMEM, including 10% fetal bovine serum and 5% carbon dioxide, at 37 °C. The BV2 cells were treated with Mdivi-1 30 min before the addition of lipopolysaccharide (LPS) at 100 ng/mL to measure the anti-inflammatory effects of Mdivi-1.

### 4.2. Animal Preparation

Male Sprague Dawley rats weighing between 350 and 450 g were used (BioLasco; Taipei; Taiwan). All rats were housed at a constant temperature (24 °C) and at regular light/dark cycles between 6:00 am and 6:00 pm, with free access to a diet. The study procedures were executed in accordance with the protocol approved by the Committee of Institutional Animal Research at the Kaohsiung Medical University (IACUC 108215).

### 4.3. Hyperglycemia Induction

Hyperglycemia was induced according to the method described by Yu-Hua Huang [44]. A single dose of STZ at 50 mg/kg was intraperitoneally injected to induce hyperglycemia. Blood glucose was measured via the tail vein of each rat using a portable glucometer (Accu-Chek Performa, Roche Diagnostics Ltd., Indianapolis, IN, USA) that was calibrated according to the manufacturer’s protocols. Diabetes mellitus (DM) induction was considered successful if the blood-glucose level was >300 mg/dL at 7 days following STZ administration [32].

### 4.4. SAH Induction

The one-shot SAH model was used in rats [45]. Briefly, the animals were anesthetized using an intraperitoneal injection of Zoletil 50^®^ (VIRBAC; Carros; France) at 40 mg/mL, which contained a mixture of zolazepam and tiletamine hydrochloride (Virbac, Carros, France). The rats’ heads were fixed in a stereotactic apparatus, and a 25-gauge butterfly needle was advanced into the cisterna magna, which was confirmed by withdrawing 0.3-mL of cerebrospinal fluid (CSF). Fresh, autologous, and non-heparinized blood (0.1 mL/100 g of body weight) drawn from the central tail artery was slowly instilled into the subarachnoid space through a butterfly needle and tubing. The animals were then kept in a ventral recumbent position for at least 30 min to promote ventral blood distribution. The respiratory pattern of rats was closely inspected, and mechanical ventilation was provided if required. Once fully awake, the animals were sent back to the vivarium.

### 4.5. Experimental Design and Drug Administration

The rats were randomly selected and divided into the following five groups (n = 6 per group): (1) control (no DM or SAH), (2) SAH only, (3) DM + SAH, (4) DM + SAH + Mdivi-1 (0.24 mg/kg), and (5) DM + SAH + Mdivi-1 (1.2 mg/kg). The dose and the time point of the Mdivi-1 treatment were chosen based on the previous study [15]. Mdivi-1 (MedChemExpress; Monmouth Junction; USA) was dissolved in 0.1% dimethyl sulfoxide (DMSO) and administered via intraperitoneal injection immediately after SAH induction. The vehicle animals received an intraperitoneal injection of 0.1% DMSO. The blood-glucose levels were monitored before and after STZ injection. The SAH model was established on day 7 after STZ injection. The animals that survived for 2 days after SAH were included for analysis, and then sacrificed for subsequent experiments.

### 4.6. Neurological Assessment

Neurobehavioral evaluation of animals was performed by assessing the sensorimotor integration of the forelimb and hindlimb activities using the modified limb-placing test, which consisted of ambulation, as well as placing and stepping reflex [46]. The motor-deficit index (MDI) represented the sum of scores for walking using the lower limbs and for placing/stepping response, and was determined before and 48 h after SAH induction. High MDI values indicated poor neurological outcomes.

### 4.7. Tissue Processing

At the end of the experiments, each animal was re-anesthetized for perfusion and fixation. The thoracic cage was opened by canalling the left ventricle using a No. 16 catheter. The brain was perfused with 180 mL of 2% paraformaldehyde and 100 mL of phosphate buffer (0.01 M) at below 36 °C and 100-mmHg perfusion pressure, after clamping the descending aorta and puncturing the right atrium. Gross inspection of harvested brains was performed to confirm the presence of subarachnoid blood clots over the BA, and the specimen was immersed in a fixative solution. The BAs were then separated from the brainstems, and the middle third of each vessel was dissected. The arterial segments were flat-embedded in paraffin, and BA cross-sections were cut into 3-μm sections that were stained with hematoxylin and eosin stain for subsequent analysis.

### 4.8. Morphometric Assessment of BA

Three cross-sections from the middle-third BA from each animal were analyzed by a trained member of research staff who was blinded to the experimental groups. The BA thickness was defined as the largest vertical distance between the inner surface of the endothelium and the outer surface of the adventitia. The arterial cross-sectional area was calculated using a computer-based morphometric analysis (ImageJ; Universal Imaging Corp., Hialeah, FL, USA). The average area of the BA cross-section from each rat was calculated to obtain the mean values for the degree of vasospasm at 48 h after SAH.

### 4.9. Immunofluorescence

After deparaffinization and rehydration, the paraffin-embedded brain samples were treated with steam heating for antigen retrieval (30 min) using a DAKO antigen retrieval solution (DAKO, Carpinteria, CA, USA). The slides were washed twice with TBS, and the sections were incubated with mouse anti-GFAP (Sigma-Aldrich; G3893; St. Louis, MI, USA) and rabbit anti-Iba1 (Proteintech; 10904-1-AP; Taiwan) antibodies for 16 h at 4 °C. The slides were, again, washed twice with TBS, and subsequently incubated with goat anti-rabbit IgG (H+L)-FAM (Croyez; C04013; Taiwan) and goat anti-mouse IgG (H+L)-TAMRA (Croyez; C04012; Taiwan) antibodies for 90 min at room temperature. Afterward, the slides were washed twice with TBS and were mounted using Fluoroshield^TM^ with DAPI (Sigma-Aldrich; F6057; St. Louis, MI, USA).

### 4.10. Enzyme-Linked Immunosorbent Assay (ELISA)

To remove the cells, the samples were immediately centrifuged at 2000× *g* for 10 min at 4 °C. The collected supernatant was stored below −15 °C before analysis. The collected body fluids were first concentrated by passing them through the C2 columns, to determine the amounts of tumor necrosis factor (TNF)-α, interleukin (IL)-1β, and IL6 in the collected supernatant (Amersham, Nutley, NJ, USA). The amount of inflammatory factor present in the media was detected using an inflammatory factor ELISA system (Thermo Fisher Scientific, Waltham, MA, USA) at 450 nm.

### 4.11. TUNEL Staining

Apoptotic nuclei were stained using a TUNEL detection kit (Roche Inc., Indianapolis, IN, USA) according to the manufacturer’s protocols. Briefly, the tissue sections were fixed in 4% methanol-free paraformaldehyde at 4 °C and washed with phosphate-buffered saline (PBS) for 30 min. Equilibrium buffer (0.1 mL) was added to each slide, which was then covered with parafilm for 10 min. A 50-μL mixture comprising 45 μL equilibrium buffer, 5-μL nucleotide mix, and 1-μL TdT enzyme was prepared and added onto each slide. The slides were incubated in the dark for 1–2 h at 37 °C. Saline sodium citrate (2×) was added to stop the TdT enzyme reaction for 15 min at room temperature. The unbound fluorescent-12-dUTP was washed out using PBS. Then, the slides were dipped in propidium iodide to stain the cells in the dark for 15 min. The slides were dried after rinsing with de-ionized water, and coverslips were overlaid on the interesting area of the slides.

### 4.12. Western Blot Analysis

Tissue extracts were prepared in 500-μL of RIPA buffer (Millipore; 20-188; Burlington, MA, USA), including 1× protease inhibitor (Roche, Germany) and 1× phosphatase inhibitor (Roche, Germany), and incubated on ice for 30 min. The protein amount in the supernatant was quantified using a BCA kit (Sigma-Aldrich; C2284; USA), and the samples (50 μg) were electrophoresed on 10% SDS–polyacrylamide gels, and then transferred to PVDF membranes following centrifugation at 13,000 rpm (10,000 g) for 30 min at 4 °C. The membranes were blocked for 60 min at room temperature in TBS containing 5% fat-free milk, and then incubated for 16 h at 4 °C with antibodies against β-actin (Sigma-Aldrich; A6316; USA) and p-Drp1 (ProSci; 79-951; Fort Collins, CO, USA). After washing, the membranes were incubated for 1.5 h at room temperature with the appropriate horseradish peroxidase-labeled secondary antibodies, and bound antibodies were visualized and quantified via chemiluminescence detection. β-actin was used as the internal control. The amount of the protein of interest, expressed as arbitrary densitometric units, was normalized to the densitometric units of β-actin; then, the density of the band was expressed as the relative density compared with that in the control group.

### 4.13. Real-Time Quantitative PCR (qRT-PCR)

The expression levels of inflammatory factors (TNF-α, IL-1β, and IL-6) were detected via qRT-PCR, using the cDNA as the template, on a StepOne Real-Time PCR System (Applied Biosystems, Waltham, MA, USA). PCR reactions were performed in triplicate, and data were analyzed using the comparative threshold cycle (2^−ΔΔCT^) method. The PCR amplification cycles consisted of denaturing at 95 °C for 5 min, 45 cycles of denaturing at 95 °C for 90 s, annealing at 61 °C for 30 s, extension at 72 °C for 30 s, and final elongation at 72 °C for 10 min. To minimize errors arising from variations in the amount of starting RNA in the samples, glyceraldehyde 3-phosphate dehydrogenase (GAPDH) was used as an internal reference. GAPDH: Forward 5′-AGACAGCCGCATCTTCTTGT-3′ and Reverse 5′-CTTGCCGTGGGTAGAGTCAT; TNF-α: Forward 5′-GCCCAGACCCTCACACTC-3′ and Reserve 5′-CACTCCAGCTGCTCCTCT-3′; IL-1β: Forward 5′- ATGGCAGAAGTACCTAAGCTCGC-3′ and Reverse 5′-ACACAAATTGCATGGTGAAGTCAGTT-3′; IL-6: Forward 5′-CCGGAGAGGAGACTTCACAG-3′ and Reverse 5′-ACAGTGCATCATCGCTGTTC-3′.

### 4.14. Statistical Analysis

The results were analyzed using Statistical Package for the Social Sciences version 20.0 (IBM SPSS Statistics). Data were expressed as mean ± standard deviation (SD). The estimates were compared with the one-way analysis of variance (ANOVA). A *p*-value of <0.05 was defined as statistically significant.

## 5. Conclusions

Our previous study revealed that hyperglycemia exacerbated cerebral vasospasm and was associated with poorer neurological outcomes following SAH. In addition, hyperglycemia enhanced inflammation and neuron cell apoptosis in the brain with SAH. Mdivi-1, a drp-1 inhibitor, attenuated cerebral vasospasm, poorer neurological outcomes, inflammation, and neuron cell apoptosis following SAH plus hyperglycemia. This study reveals that hyperglycemia enhanced SAH-induced Drp1 phosphorylation. Mdivi-1, a drp-1 inhibitor, attenuated cerebral vasospasm, poor neurological outcomes, inflammation, and neuron cell apoptosis following SAH, with or without hyperglycemia. The data suggest that hyperglycemia aggravated SAH-induced neurological prognosis through mitochondrial fission. Therefore, after Mdivi-1 treatment in hyperglycemic SAH animals, the inhibition of Drp1 significantly reduces the morphological change in mitochondria while alleviating neurobehavioral deficits. The number of apoptotic neurons decreased in a dose-dependent manner. The results prove that Mdivi-1 has therapeutic effects against hyperglycemia-exacerbated neuronal death.

## Figures and Tables

**Figure 1 ijms-23-06924-f001:**
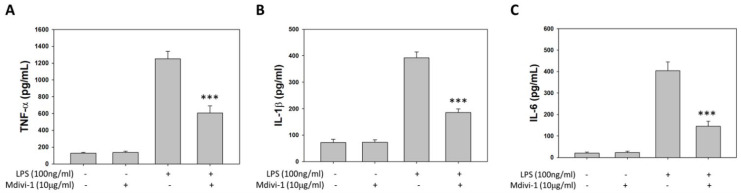
ELISA assay pro-inflammatory factors in BV2 cells after LPS induction using Mdivi-1. The levels of (**A**) TNF-α, (**B**) IL-1β, and (**C**) IL-6 were measured using commercially available kits. All values are expressed as mean ± SD (n = 6). *** *p* < 0.001 compared with induced LPS.

**Figure 2 ijms-23-06924-f002:**
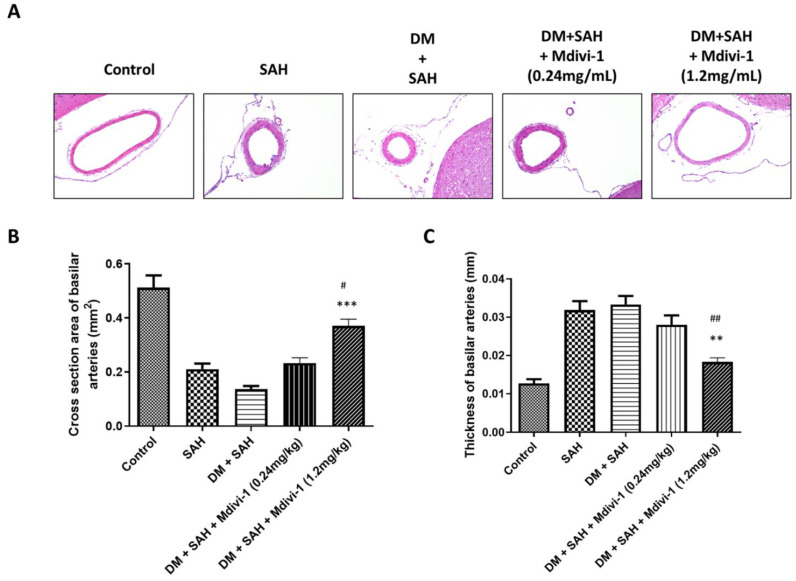
Representative micrographs of BA cross-sections obtained from (**A**) the control, SAH, DM + SAH, DM + SAH + Mdivi-1 (0.24 mg/kg), and DM + SAH + Mdivi-1 (1.2 mg/kg) groups; (**B**) comparison of the BA cross-sectional area among the control, SAH, DM + SAH, DM +SAH + Mdivi-1 (0.24 mg/kg), and DM + SAH + Mdivi-1 (1.2 mg/kg) groups; (**C**) comparison of the thickness of BA among the same five groups. All values are expressed as mean ± SD (n = 6). ^#^
*p* < 0.05, ^##^
*p* < 0.01 compared with the SAH group. ** *p* < 0.01, *** *p* < 0.001 compared with the DM + SAH group.

**Figure 3 ijms-23-06924-f003:**
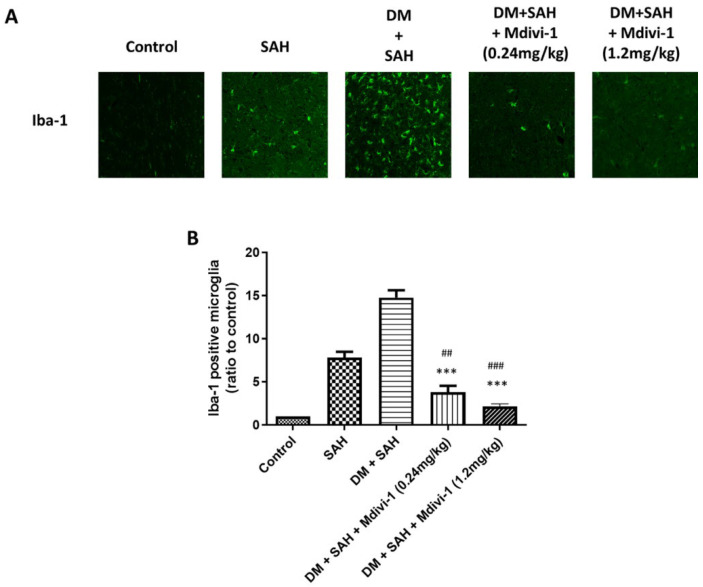
Microglial proliferation in the rat brain as determined via immunofluorescence staining for Iba-1: (**A**) Representative micrographs of Iba-1 staining are shown for the control, SAH, DM + SAH, DM + SAH + Mdivi-1 (0.24 mg/kg), and DM + SAH + Mdivi-1 (1.2 mg/kg) groups; and (**B**) the intensities of immunofluorescence staining in the images were quantified relative to the levels of the control animals. All values are expressed as mean ± SD (n = 6). ^##^
*p* < 0.01, ^###^
*p* < 0.001 compared with the SAH group. *** *p* < 0.001 compared with the DM + SAH group.

**Figure 4 ijms-23-06924-f004:**
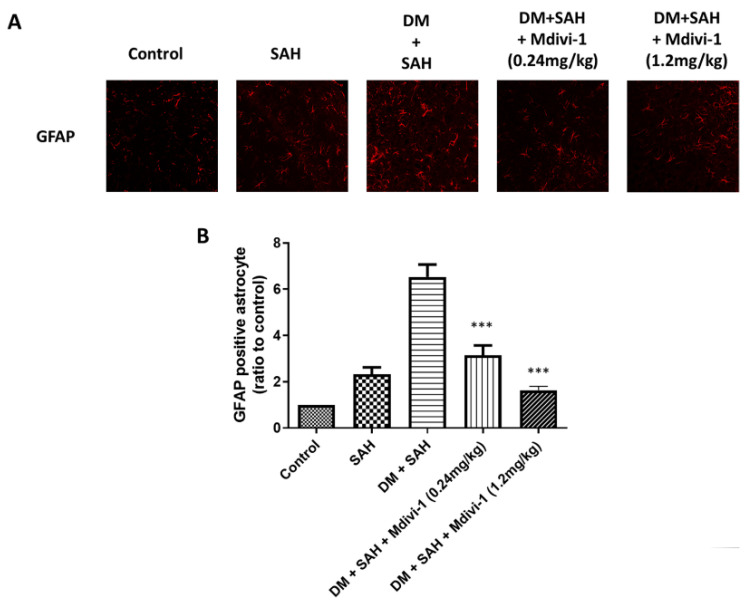
Astrocyte proliferation in the rat brain as determined via immunofluorescence staining for GFAP: (**A**) Representative micrographs of GFAP staining are shown for the control, SAH, DM + SAH, DM + SAH + Mdivi-1 (0.24 mg/kg), and DM + SAH + Mdivi-1 (1.2 mg/kg) groups; and (**B**) the immunofluorescence staining intensities in the images were quantified relative to the levels of the control animals. All values are expressed as mean ± SD (n = 6). *** *p* < 0.001 compared with the DM + SAH group.

**Figure 5 ijms-23-06924-f005:**
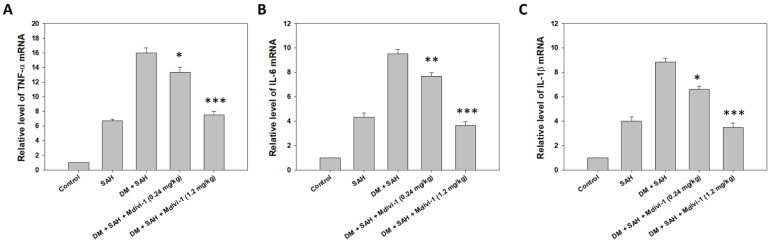
Real-time PCR for pro-inflammatory factors in the brains of rats following SAH. The samples were collected 7 days after SAH. The (**A**) TNF-α, (**B**) IL-1β, and (**C**) IL-6 levels were measured using commercially available kits. All values are expressed as mean ± SD (n = 6). * *p* < 0.05, ** *p* < 0.01, *** *p* < 0.001 compared with the DM + SAH group.

**Figure 6 ijms-23-06924-f006:**
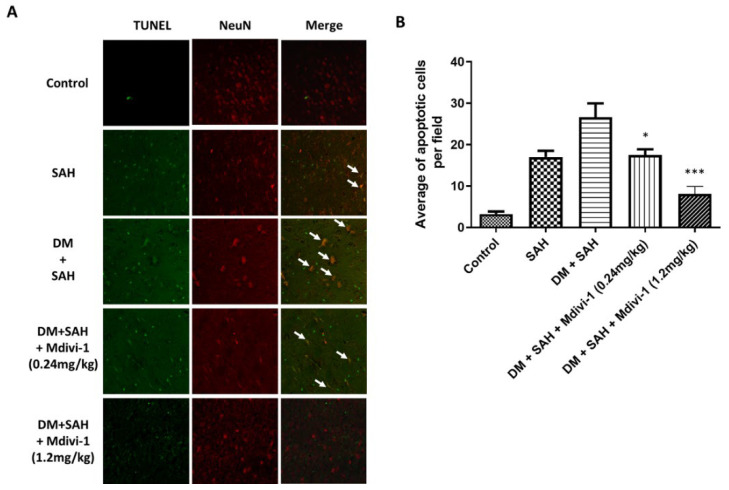
Neuron cell apoptosis in the rat brain as determined via immunofluorescence staining for TUNEL and NeuN: (**A**) Representative micrographs of immunofluorescence staining for TUNEL and NeuN are shown for the control, SAH, DM + SAH, DM + SAH + Mdivi-1 (0.24 mg/kg), and DM + SAH + Mdivi-1 (1.2 mg/kg) groups; and (**B**) the number of instances of neuron cell apoptosis from immunofluorescence staining in the images was measured in all groups. All values are expressed as mean ± SD (n = 6). * *p* < 0.05, *** *p* < 0.001 compared with the DM + SAH group.

**Figure 7 ijms-23-06924-f007:**
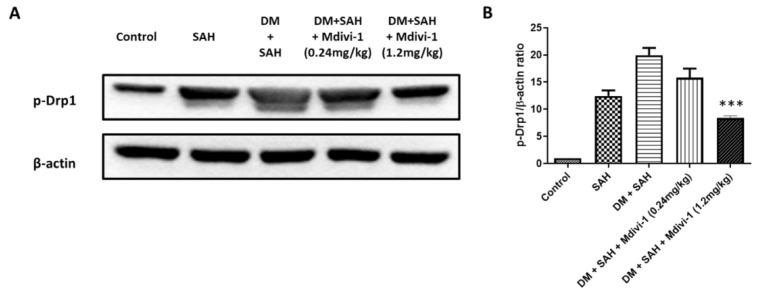
Western blot analysis showing the effect of the Mdivi-1 treatment on the levels of p-Drp1 protein levels in the cortex of rats at 7 days after SAH: (**A**) Representative results of Western blot analyses; (**B**) the p-Drp1 expression levels were normalized in the control group. All values are expressed as mean ± SD (n = 6). *** *p* < 0.001 compared with the DM + SAH group.

**Table 1 ijms-23-06924-t001:** Behavioral assessment.

Group	Ambulation	Placing/Stepping Reflex	MDI
Normal	0	0	0
SAH	1.91 ± 0.37	1.34 ± 0.11	2.86 ± 0.24
DM + SAH	2.39 ± 0.27	1.77 ± 0.19	4.20 ± 0.31
DM + SAH + Mdivi-1 (0.24 mg/kg)	2.11 ± 0.19	1.59 ± 0.29	3.83 ± 0.25
DM + SAH + Mdivi-1 (1.2 mg/kg)	1.62 ± 0.13 *	0.57 ± 0.21 *	1.89 ± 0.31 *

MDI—motor-deficit index; SAH—subarachnoid hemorrhage. Data are expressed as the mean ± standard error of mean (SEM, n = 6/group); * *p* < 0.05 vs. DM + SAH, via Mann–Whitney U test.

## Data Availability

Not applicable.

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
