# Peer review of "Therapeutic Effect of Mitochondrial Division Inhibitor-1 (Mdivi-1) on Hyperglycemia-Exacerbated Early and Delayed Brain Injuries after Experimental Subarachnoid Hemorrhage"

_ijms, 2022, doi:10.3390/ijms23136924_

Round 1

Reviewer 1 Report

This study investigated the effect of Mdivi-1, a Drp1 inhibitor, on hyperglycemia-exacerbated brain injuries after experimental SAH, focusing on the possibility of treating the worst neurological prognosis of SAH related to hyperglycemia by modulating mitochondrial dynamics.

The topic, although interesting has not been adequately addressed, showing several critical points: the description of the results is overall lacking, excessively concise, in some cases, incorrect, in one case even missing (paragraph 2.2. of Results), therefore, the manuscript requiring a major revision.

Major points:

Paragraph 2.1 ELISA assay for inflammatory factors in vitro

The assay of cytokines by ELISA is described, generically indicating "supernatant samples" without specifying which cells are referred to

The Paragraph 2.2 "Neurological outcomes" does not report the relative results but the repetition of the legend of Figure 1

The results of the Table 1 are not described, probably because they should have been included in paragraph 2.2 "Neurological outcomes" which, as already mentioned, does not report the data relating to "Neurological outcomes"

Paragraph 2.3 Morphological, Cross-sectional area, and thickness changes in BA

The paragraph contains the acronym BA without reporting the full name. The paragraph has been placed in the middle and separates the title of Table 1 "Table1. Behavioral assessment" and the table 1 that has been inserted after paragraph 2.3.

Paragraph 2.4 Proliferation of microglia and astrocytes

Lines 164-166: “Quantitative analysis of the intensity of Iba-1 staining also revealed comparable levels between the control (set at 1.0), SAH (7.80 ± 165 1.88), DM + SAH (14.76 ± 2.07), DM + SAH + low dose Mdivi-1 (0.24mg/kg) (3.81 ± 1.80),  and DM + SAH + high dose Mdivi-1 (1.2mg/kg) groups (2.15 ± 0.74).”

How can the levels of microglia proliferation be considered "comparable" by matching values of 1.0 (set in the control) with 7.8, 14,76, 3.81 and 2.15?

Lines 174-176 : “Quantitative analysis of the intensity of 173 GFAP staining also revealed comparable levels between the control (set at 1.0), SAH 174 (2.33 ± 0.72), DM + SAH (6.53 ± 1.32), DM + SAH + low dose Mdivi-1 (0.24mg/kg) (3.14 ± 175 1.63), and DM + SAH + high dose Mdivi-1 (1.2mg/kg) groups (1.63 ± 0.42)”

Similarly, how can the level of astrocyte proliferation be considered “comparable” by matching values of 1.0 (set in the control) with 2,33, 6,53, 3,14, 1,63?

The paragraph of the conclusions is telegraphic and too concise

Minor

Abstract: The acronyms LPS and DM reported for the first time in the abstract (as in the results) are indicated without reporting the full name of what they indicate

Figure 7:  there is disproportion between the dimensions of representative results of Western blots shown in Figure 7A (too large) with respect to the histogram of the levels of pDrp1 shown in Figure 7B

Author Response

This study investigated the effect of Mdivi-1, a Drp1 inhibitor, on hyperglycemia-exacerbated brain injuries after experimental SAH, focusing on the possibility of treating the worst neurological prognosis of SAH related to hyperglycemia by modulating mitochondrial dynamics.

The topic, although interesting has not been adequately addressed, showing several critical points: the description of the results is overall lacking, excessively concise, in some cases, incorrect, in one case even missing (paragraph 2.2. of Results), therefore, the manuscript requiring a major revision.

Major points:

Paragraph 2.1 ELISA assay for inflammatory factors in vitro

The assay of cytokines by ELISA is described, generically indicating "supernatant samples" without specifying which cells are referred to

Thanks for your valuable recommend. The “supernatant samples” means the supernatant from BV-2. Therefore, in Line 128-129, we modified “The TNF-α, IL-1β, and IL-6 levels in the supernatant samples from BV-2 were examined using ELISA 6 h after LPS induction to…..”

The Paragraph 2.2 "Neurological outcomes" does not report the relative results but the repetition of the legend of Figure 1

The results of the Table 1 are not described, probably because they should have been included in paragraph 2.2 "Neurological outcomes" which, as already mentioned, does not report the data relating to "Neurological outcomes"

Thanks for your valuable recommend. It is my mistake. We had added this result in Line 143-155. “The neurobehavioral scores, including ambulation, placing/stepping reflex, and MDI, were not different between the SAH, DM + SAH, and DM + SAH + low -dose Mdivi-1 (0.24 mg/kg) only groups (Table 1). In animals subjected to SAH, bothBoth the ambulation (1.91 ± 0.37) and placing/stepping reflex (1.34 ± 0.11) scores were significantly lower in the SAH animals than in the DM + SAH animals (ambulation: 2.39 ± 0.27 and placing/stepping reflex: 1.77 ± 0.19). Treatment with a high- dose of Mdivi-1 (1.2 mg/kg) significantly decreased both the ambulation (1.62 ± 0.13; P < 0.05) and the placing/stepping reflex (0.57 ± 0.21) scores when compared with the DM + SAH group, but treatment withbut not the low -dose Mdivi-1 (0.24 mg/kg) did not treatment (ambulation: 2.11 ± 0.19 and placing/stepping reflex: 1.59 ± 0.29). Likewise, similarly, the MDI in the high -dose Mdivi-1 (1.2 mg/kg) treatment group (1.89 ± 0.31; P < 0.05) was also significantly reduced when compared with that in the DM + SAH group (4.20 ± 0.31) (Table 1).”

Paragraph 2.3 Morphological, Cross-sectional area, and thickness changes in BA

The paragraph contains the acronym BA without reporting the full name. The paragraph has been placed in the middle and separates the title of Table 1 "Table1. Behavioral assessment" and the table 1 that has been inserted after paragraph 2.3.

Thanks for your valuable recommend. We had added full name in paragraph 2.3 “Morphological, Crosscross-sectional area, and thickness changes in basal artery (BA)”

Paragraph 2.4 Proliferation of microglia and astrocytes

Lines 164-166: “Quantitative analysis of the intensity of Iba-1 staining also revealed comparable levels between the control (set at 1.0), SAH (7.80 ± 165 1.88), DM + SAH (14.76 ± 2.07), DM + SAH + low dose Mdivi-1 (0.24mg/kg) (3.81 ± 1.80),  and DM + SAH + high dose Mdivi-1 (1.2mg/kg) groups (2.15 ± 0.74).”

How can the levels of microglia proliferation be considered "comparable" by matching values of 1.0 (set in the control) with 7.814,763.81 and 2.15?

Thanks for your valuable recommend. We considered the studies “2-PMAP Ameliorates Cerebral Vasospasm and Brain Injury after Subarachnoid Hemorrhage by Regulating Neuro-Inflammation in Rats”, “Blocking Hepatoma-Derived Growth Factor Attenuates Vasospasm and Neuron Cell Apoptosis in Rats Subjected to Subarachnoid Hemorrhage”, and “Attenuation in Proinflammatory Factors and Reduction in Neuronal Cell Apoptosis and Cerebral Vasospasm by Minocycline during Early Phase after Subarachnoid Hemorrhage in the Rat”. In addition, we evaluated each group at the same conditions including the concentration of antibody, the value of gain and seconds in photos. 

Lines 174-176 : “Quantitative analysis of the intensity of 173 GFAP staining also revealed comparable levels between the control (set at 1.0), SAH 174 (2.33 ± 0.72), DM + SAH (6.53 ± 1.32), DM + SAH + low dose Mdivi-1 (0.24mg/kg) (3.14 ± 175 1.63), and DM + SAH + high dose Mdivi-1 (1.2mg/kg) groups (1.63 ± 0.42)”

Similarly, how can the level of astrocyte proliferation be considered “comparable” by matching values of 1.0 (set in the control) with 2,336,533,141,63?

The paragraph of the conclusions is telegraphic and too concise

Thanks for your valuable recommend. It is the same as microglia. We considered the studies “2-PMAP Ameliorates Cerebral Vasospasm and Brain Injury after Subarachnoid Hemorrhage by Regulating Neuro-Inflammation in Rats”, “Blocking Hepatoma-Derived Growth Factor Attenuates Vasospasm and Neuron Cell Apoptosis in Rats Subjected to Subarachnoid Hemorrhage”, and “Attenuation in Proinflammatory Factors and Reduction in Neuronal Cell Apoptosis and Cerebral Vasospasm by Minocycline during Early Phase after Subarachnoid Hemorrhage in the Rat”. In addition, we evaluated each group at the same conditions including the concentration of antibody, the value of gain and seconds in photos.

Minor

Abstract: The acronyms LPS and DM reported for the first time in the abstract (as in the results) are indicated without reporting the full name of what they indicate

Thanks for your valuable recommend. We had added full name in the abstract.

Figure 7:  there is disproportion between the dimensions of representative results of Western blots shown in Figure 7A (too large) with respect to the histogram of the levels of pDrp1 shown in Figure 7B

Thanks for your valuable recommend. We replaced Figure 7A with other data.

Reviewer 2 Report

The English needs quite a bit of work. The Results are incompetently presented. In one section the text has been forgotten. Statistics have been done inappropriately, inadequately.

Author Response

The English needs quite a bit of work.

Thanks for your valuable recommend. We had asked professionals to revise the English grammar.

The Results are incompetently presented. In one section the text has been forgotten.

Thanks for your valuable recommend. It is my mistake. We had added this result in Line 143-155. “The neurobehavioral scores, including ambulation, placing/stepping reflex, and MDI, were not different between the SAH, DM + SAH, and DM + SAH + low -dose Mdivi-1 (0.24 mg/kg) only groups (Table 1). In animals subjected to SAH, bothBoth the ambulation (1.91 ± 0.37) and placing/stepping reflex (1.34 ± 0.11) scores were significantly lower in the SAH animals than in the DM + SAH animals (ambulation: 2.39 ± 0.27 and placing/stepping reflex: 1.77 ± 0.19). Treatment with a high- dose of Mdivi-1 (1.2 mg/kg) significantly decreased both the ambulation (1.62 ± 0.13; P < 0.05) and the placing/stepping reflex (0.57 ± 0.21) scores when compared with the DM + SAH group, but treatment withbut not the low -dose Mdivi-1 (0.24 mg/kg) did not treatment (ambulation: 2.11 ± 0.19 and placing/stepping reflex: 1.59 ± 0.29). Likewise, similarly, the MDI in the high -dose Mdivi-1 (1.2 mg/kg) treatment group (1.89 ± 0.31; P < 0.05) was also significantly reduced when compared with that in the DM + SAH group (4.20 ± 0.31) (Table 1).”

Statistics have been done inappropriately, inadequately.

Thanks for your valuable recommend. I had asked statistician in Kaohsiung medical University to renew statistics in this study.

Round 2

Reviewer 1 Report

The authors have appropriately modified the manuscript according to the suggestions of the reviewer, who considers the modified version of the manuscript acceptable for publication in its current form.

Note and correct the error in inserting Table 1: bring forward the insertion of Table 1 after paragraph 2.2, currently it has been inserted after paragraph 2.3